# The Relationship between the IC_50_ Values and the Apparent Inhibition Constant in the Study of Inhibitors of Tyrosinase Diphenolase Activity Helps Confirm the Mechanism of Inhibition

**DOI:** 10.3390/molecules27103141

**Published:** 2022-05-13

**Authors:** Pablo Garcia-Molina, Francisco Garcia-Molina, Jose Antonio Teruel-Puche, Jose Neptuno Rodriguez-Lopez, Francisco Garcia-Canovas, Jose Luis Muñoz-Muñoz

**Affiliations:** 1GENZ-Group of Research on Enzymology, Department of Biochemistry and Molecular Biology-A, Regional Campus of International Excellence “Campus Mare Nostrum”, University of Murcia, Espinardo, 30100 Murcia, Spain; pablo.garcia14@um.es (P.G.-M.); neptuno@um.es (J.N.R.-L.); canovasf@um.es (F.G.-C.); 2Department of Anatomía Patológica, Hospital General Universitario Reina Sofía, Av. Intendente Jorge Palacios, 1, 30003 Murcia, Spain; 3Department of Biochemistry and Molecular Biology-A, Regional Campus of International Excellence “Campus Mare Nostrum”, University of Murcia, Espinardo, 30100 Murcia, Spain; teruel@um.es; 4Microbial Enzymology Lab, Department of Applied Sciences, Ellison Building A, University of Northumbria, Newcastle Upon Tyne NE1 8ST, UK

**Keywords:** tyrosinase, polyphenol oxidase, diphenolase activity, IC_50_, inhibition, $K_{I}^{\mathrm{app}}$

## Abstract

Tyrosinase is the enzyme involved in melanization and is also responsible for the browning of fruits and vegetables. Control of its activity can be carried out using inhibitors, which is interesting in terms of quantitatively understanding the action of these regulators. In the study of the inhibition of the diphenolase activity of tyrosinase, it is intriguing to know the strength and type of inhibition. The strength is indicated by the value of the inhibition constant(s), and the type can be, in a first approximation: competitive, non-competitive, uncompetitive and mixed. In this work, it is proposed to calculate the degree of inhibition ($i_{D}$), varying the concentration of inhibitor to a fixed concentration of substrate, L-dopa (D). The non-linear regression adjustment of $i_{D}$ with respect to the initial inhibitor concentration ${\left[I\right]}_{0}$ allows for the calculation of the inhibitor concentration necessary to inhibit the activity by 50%, at a given substrate concentration (IC_50_), thus avoiding making interpolations between different values of $i_{D}$. The analytical expression of the IC_50_, for the different types of inhibition, are related to the apparent inhibition constant ($K_{I}^{\mathrm{app}})$. Therefore, this parameter can be used: (a) To classify a series of inhibitors of an enzyme by their power. Determining these values at a fixed substrate concentration, the lower IC_50_, the more potent the inhibitor. (b) Checking an inhibitor for which the type and the inhibition constant have been determined (using the usual methods), must confirm the IC_50_ value according to the corresponding analytical expression. (c) The type and strength of an inhibitor can be analysed from the study of the variation in $i_{D}$ and IC_50_ with substrate concentration. The dependence of IC_50_ on the substrate concentration allows us to distinguish between non-competitive inhibition ($i_{D}$ does not depend on ${\left[D\right]}_{0}$) and the rest. In the case of competitive inhibition, this dependence of $i_{D}$ on ${\left[D\right]}_{0}$ leads to an ambiguity between competitive inhibition and type 1 mixed inhibition. This is solved by adjusting the data to the possible equations; in the case of a competitive inhibitor, the calculation of $K_{I_{1}}^{\mathrm{app}}$ is carried out from the IC_50_ expression. The same occurs with uncompetitive inhibition and type 2 mixed inhibition. The representation of $i_{D}$ vs. n, with $n={\left[D\right]}_{0}/K_{m}^{D}$, allows us to distinguish between them. A hyperbolic $i_{D}$ vs. n representation that passes through the origin of coordinates is a characteristic of uncompetitive inhibition; the calculation of $K_{I_{2}}^{\mathrm{app}}$ is immediate from the IC_50_ value. In the case of mixed inhibitors, the values of the apparent inhibition constant of meta-tyrosinase (Em) and oxy-tyrosinase (Eox), $K_{I_{1}}^{\mathrm{app}}$ and the apparent inhibition constant of metatyrosinase/Dopa complexes (EmD) and oxytyrosinase/Dopa (EoxD), $K_{I_{2}}^{\mathrm{app}}$ are obtained from the dependence of $i_{D}$ vs. n, and the results obtained must comply with the IC_50_ value.

## 1. Introduction

Tyrosinase (EC 1.14.18.1) is an enzyme widely distributed in nature, including in bacteria, fungi, plants and animals [1]. The enzyme catalyses the rate-limiting step in melanin biosynthesis in mammals, the conversion of L-tyrosine to L-dopa and the oxidation of the latter to L-*o*-dopaquinone [2] (Figure 1). The abnormal functioning of tyrosinase causes hyperpigmentation or hypopigmentation phenomena [3,4].

There is great interest in the control of tyrosinase activity, especially through the use of inhibitors, which can be natural or synthetic [5,6,7,8,9,10,11]. This enzyme has two activities: firstly, monophenolase activity, catalysing the passage from L-tyrosine to L-dopa, and secondly, diphenolase activity, which catalyses the passage from L-dopa to L-*o*-dopaquinone [1,2]. Monophenolase activity exhibits a lag period τ before reaching steady state [12]. For this reason, the study of inhibitors on this activity of the enzyme is difficult and it is best to eliminate the delay period by adding a certain amount of L-dopa, necessary to reach the steady state. Once this addition is made, tyrosinase behaves kinetically as a Michaelian enzyme [13].

Diphenolase activity is of the Michaelian type and is normally used to study the kinetics of inhibitors, characterizing the type of inhibition and the strength of the inhibitor [13]. In addition, a parameter is calculated, the IC_50_, which indicates the concentration of inhibitor that causes 50% inhibition under certain experimental conditions [14,15]. This IC_50_ parameter is related to the value of the apparent inhibition constant, $K_{I}^{\mathrm{app}}$ [16].

The diagnostic value of the reversible enzymatic inhibition of the IC_50_ parameter and its relationship with $K_{I}^{\mathrm{app}}$ has been described for some time [16]. In the case of a monosubstrate reaction and under fast equilibrium conditions, the relationship between IC_50_ and $K_{I}^{\mathrm{app}}$ (competitive, non-competitive and uncompetitive inhibitors) has been established. In these analytical expressions, the dependence of $i_{D}$ (degree of inhibition of diphenolase activity) on concentration of substrate on which the experiments are performed is revealed, except in the case of non-competitive inhibition [16]. An analysis of data, based on the plots of $1/{\mathrm{IC}}_{50}$ vs. $V_{0}/V_{\max}$, the unihibited rate ($V_{0}$) divided by the maximal rate ($V_{\max}$), was proposed [17]. Subsequently, making use of the dependence of the inhibition degree on the ratio of the substrate concentration, ${\left[S\right]}_{0}$, to the Michaelis constant ($K_{m}^{S}$), ${\left[S\right]}_{0}/K_{m}^{S}$ discrimination was addressed between mechanisms, considering a monosubstrate reaction in rapid equilibrium [18]. Experimental design and data analysis based on the dependence of IC_50_ vs. ${\left[S\right]}_{0}/K_{m}$ has recently been proposed [19].

Since tyrosinase inhibitors can control the activity of the enzyme, their study is very important. Thus, in mammalian pigmentation, an excessive action of tyrosinase causes hyperpigmentation, such as melasma, freckles and ephelides. In addition, the enzyme is responsible for the browning of fruits, vegetables, fungi and crustaceans, which leads to a decrease in commercial value. For these reasons, we focus our study on the quantitative characterization of the inhibitors. The purpose of this work is to establish quantitative relationships between $i_{D}$ and the inhibitor concentration ${\left[I\right]}_{0}$ at a fixed substrate concentration, and with the relationship ${\left[D\right]}_{0}/K_{m}=n$ at a fixed concentration of inhibitor, where ${\left[D\right]}_{0}$ is the initial concentration of L-dopa, for the diphenolase activity of tyrosinase. From these data it is possible to determine, from the dependence of $i_{D}$ vs. ${\left[I\right]}_{0}$, the value of IC_50_. This value should make it possible to: (a) Order different inhibitors by their inhibitory power, where for a lower IC_50_, the more powerful the inhibitor is; (b) Check an inhibitor studied via the usual methods ($1/V_{0,\;i}$ vs. $1/{\left[D\right]}_{0}$ at different concentrations of ${\left[I\right]}_{0}$) whose type and strength are known ($K_{I}^{\mathrm{app}}$). It must be confirmed that the analytical expression of the IC_50_ corresponding to this type of inhibition is fulfilled; (c) Analyse the variation in $i_{D}$ and IC_50_ value with respect to the substrate concentration. If when the value of ${\left[D\right]}_{0}$ varies, $i_{D}$ does not vary, the inhibition is non-competitive and the value of $K_{I}^{\mathrm{app}}$ is equal to the value of IC_50_. In other cases, to determine the type and strength of the inhibitor, the variation of $i_{D}$ vs. ${\left[D\right]}_{0}$ must be studied. The experimental results obtained must comply with the analytical expressions deduced for IC_50_. In turn, this confirms the validity of the kinetic study carried out.

## 2. Results and Discussion

### 2.1. Diphenolase Activity

Tyrosinase catalyses the hydroxylation of monophenols to *o*-diphenols (monophenolase activity) and the oxidation of *o*-diphenols to *o*-quinones (diphenolase activity).

In the case of tyrosinase diphenolase activity, the mechanism is in Figure 2 [20]:

The general case of the inhibition of this activity can be expressed according to Figure 3:

From the mechanism described in Figure 3, different particular cases can be considered: competitive inhibition, where the inhibitor binds preferentially to the enzymatic forms Em and Eox in such a way that $K_{I_{2}}^{\mathrm{app}}\to \infty$; non-competitive inhibition, where the inhibitor binds to the free enzymatic forms (Em and Eox) and to the complexes (EmD and EoxD), fulfilling $K_{I_{1}}^{\mathrm{app}}≅K_{I_{2}}^{\mathrm{app}}$; uncompetitive inhibitors bind preferentially to the EmD complexes and EoxD such that $K_{I_{1}}^{\mathrm{app}}\to \infty$; finally, in the mixed inhibition, it is true that $K_{I_{1}}^{\mathrm{app}}\neq K_{I_{2}}^{\mathrm{app}}$ (the expressions of $K_{I_{1}}^{\mathrm{app}}$ and $K_{I_{2}}^{\mathrm{app}}$ are described in the Appendix A). Note that these constants are apparent since their analytical expressions include substrate catalysis rate constants and inhibition equilibrium constants.

The affinity of the enzyme for oxygen is so great [21,22] that at the concentration of oxygen in the solution it is saturated, and therefore the inhibitor does not bind to deoxytyrosinase (Ed) $K_{13}\to \infty$.

### 2.2. Tyrosinase Inhibition by Benzoate and Cinnamate

The inhibition of tyrosinase with benzoate and cinnamate has been studied. Working experimentally at a fixed inhibitor concentration and varying the substrate concentration, the initial velocities ($V_{o,i}^{D,\mathrm{DC}}$) and the representation of $1/V_{o,i}^{D,\mathrm{DC}}$ vs. $1/{\left[D\right]}_{0}$ gives a straight line. This approach is repeated at various inhibitor concentrations (the lines intersect at the $1/V_{o,i}^{D,\mathrm{DC}}$ axis) and $K_{m}^{\mathrm{app}}$ is then determined at each inhibitor concentration [13]. A secondary plot of $K_{m}^{\mathrm{app}}\;\mathrm{vs}.{\left[I\right]}_{0}$ allows for determination of the apparent inhibition constant $K_{I}^{\mathrm{app}}$ [23,24,25]. The application of this methodology allows us to determine the type of inhibition and the strength of these inhibitors, meaning in this case that these compounds behave as competitive inhibitors, being benzoate and cinnamate at pH = 7.0, with $K_{I}^{\mathrm{app}}$ values of 0.53 ± 0.06 mM (benzoate) and 0.44 ± 0.04 mM (cinnamate).

#### Check of Inhibition by Cinnamate and Benzoate. Calculation of the IC_50_ Value and Its Relationship with $K_{I}^{\mathrm{app}}$

In this work, the experimental values of $i_{D}$ have been calculated at a fixed concentration of substrate, ${\left[D\right]}_{0}=K_{m}^{D}$ and different concentrations of inhibitor, both in the presence of benzoate and in the presence of cinnamate. Adjusting the values of $i_{D}$ to Appendix A, the IC_50_ values shown in Figure 1, ${\mathrm{IC}}_{50}^{B}$ for benzoate and ${\mathrm{IC}}_{50}^{C}$ for cinnamate are obtained.

With the values of ${\mathrm{IC}}_{50}^{B}=0.99\pm 0.02\;\mathrm{mM}$ and ${\mathrm{IC}}_{50}^{C}=0.80\pm 0.02\;\mathrm{mM}$ obtained for benzoate and cinnamate, respectively, so being ${\mathrm{IC}}_{50}^{C}<{\mathrm{IC}}_{50}^{B}$, then cinnamate is a more potent inhibitor than benzoate. The values of $K_{I}^{\mathrm{app}}$ (Appendix A) are calculated and compared with the values obtained from $K_{I}^{\mathrm{app}}$ following the experimental method described above for competitive inhibitors (see Table 1) [13]. Note the similarity between the values obtained by the two methods. The chemical structures of benzoate and cinnamate are shown in Figure 2. The docking of these compounds to met-tyrosinase and oxy-tyrosinase is shown in Figure 3 and Figure 4. The values of the dissociation constants are shown in Table 2. Note that at the working pH, these compounds are as benzoate and cinnamate and therefore bind better to met-tyrosinase than to oxy-tyrosinase, as shown in Figure 3 and Figure 4. The K_D_ value is higher for the case of oxy-tyrosinase (Table 2) due to the repulsion between the negative charges of the compounds and the peroxide of oxy-tyrosinase.

The chemical structures of these compounds and their derived analogous are shown in Figure 2.

The dockings of benzoate and cinnamate to oxy-tyrosinase and met-tyrosinase are shown in Figure 3 and Figure 4.

### 2.3. Discussion about the Relationship of IC_50_ and the Values of the Inhibition Constant $K_{I}^{app}$ with Data from the Literature for Tyrosinase

While conducting a literature review of recently published tyrosinase inhibition studies, we sometimes observed that there was a lack of correlation between the IC_50_ values and the $K_{I}^{\mathrm{app}}$ values. Next, some cases are described for the different types of inhibition.

#### 2.3.1. With Reference to Competitive Inhibitors

The study of tyrosinase inhibition can be carried out on diphenolase, monophenolase activity or both. When the study is carried out using L-tyrosine as a substrate (monophenolase activity), the existence of a transition phase must be taken into account, which causes a delay period, τ, before the system reaches a steady state [12]. In a previous work, our group showed the usefulness of adding, at t = 0, the necessary amount of L-dopa to reach the steady state and to be able to obtain correct measurements of the initial velocity [13].

Benzimidazothiazolone derivatives are tyrosinase inhibitors [33]. An inhibition study was carried out using L-tyrosine as substrate and the values obtained for the compounds are shown in Appendix A. It was established that the three compounds a–c are competitive inhibitors; the inhibition constants and the IC_50_ described are: (a) $K_{I}^{\mathrm{app}}=16.55{\;\mu M\;\mathrm{IC}}_{50}=3.70\pm 0.51\;\mu M$; (b) $K_{I}^{\mathrm{app}}=3.21{\;\mu M\;\mathrm{IC}}_{50}=3.05\pm 0.95\;\mu M$ and (c) $K_{I}^{\mathrm{app}}=3.01{\;\mu M\;\mathrm{IC}}_{50}=5.00\pm 0.38\;\mu M$. From these values it becomes clear for compounds (a) and (b) that $K_{I}^{\mathrm{app}}>{\mathrm{IC}}_{50}$, which is not in agreement with Appendix A. In addition, the chemical structure of the compounds described in Appendix A shows that these molecules could behave as alternative substrates to L-tyrosine, and this would also lead to a distortion in the initial velocity measurements and therefore of $K_{I}^{\mathrm{app}}$ and IC_50_. In the presence of L-dopa accumulated in the medium when the enzyme acts on L-tyrosine, these compounds can behave as alternative substrates to L-tyrosine. Regarding the docking, the compounds bind with practically the same affinity to the met and oxy forms (Table 3). The docking is shown in Appendix A. In addition, it should be noted that the distance of the oxygen from the peroxide group in the oxy-tyrosinase form is less than 2.9 Ᾰ for the ortho position of the phenolic hydroxyl of the compound, in such a way that the hydroxylation reaction can occur and behave as an alternative substrate; this could be the origin of the kinetic deviations. On the other hand, compound b, due to its ortho-diphenolic structure, can be a substrate for the enzyme, as shown in Appendix A in the docking to oxy and met-tyrosinase (Table 3). Note that when using L-tyrosine as a substrate there is a delay period, and if it is not eliminated, it can lead to erroneous initial velocity measurements [13].

Appendix A shows how a monophenol can behave as an alternative substrate to L-dopa. The met-tyrosinase form is inactive on the monophenol, but the oxy-tyrosinase form is capable of hydroxylating it if certain requirements are met. On the other hand, the met-tyrosinase form acting on L-dopa (measured substrate) enters the catalytic cycle.

In the study of urolithin and reduced urolithin derivatives as potent inhibitors of tyrosinase [26], the compounds described in Appendix A show competitive inhibition with IC_50_ values of (a) 18.09 ± 0.25 µM, (b) 4.14 ± 0.10 and (c) 15.69 µM. The values of $K_{I}^{\mathrm{app}}$ described are: 2 µM, 0.4 µM and 3 µM. In this case, it is true that IC_50_
> $K_{I}^{\mathrm{app}}$, but these values imply, according to Appendix A, ${\mathrm{IC}}_{50}=K_{I}^{\mathrm{app}}\left(1+n\right)$, which has been worked out at values of $n={\left[D\right]}_{0}/K_{m}^{D}$ of 8, 10 and 4. In addition, the chemical structure of the three compounds indicates that they could be substrates of the enzyme (Appendix A), and the docking data are shown in Table 2 and Table 3. Appendix A show the docking to tyrosinase of these compounds. The compounds a (Appendix A) and b (Appendix A) from Appendix A could be alternative substrates (Table 3). Compound c (Appendix A) from Appendix A is shown as a true competitive inhibitor (Table 2).

The study of cinnamate derivatives on tyrosinase monophenolase and diphenolase activities suggests that trans-3,4-difluorocinnamic acid (Appendix A) behaves as a competitive inhibitor (Table 2), with a value of $K_{I}^{\mathrm{app}}$= 197 ± 11 µM and an IC_50_ value of 0.78 ± 0.02 mM, which approximately satisfies the Appendix A
${\mathrm{IC}}_{50}=K_{I}^{\mathrm{app}}\left(1+n\right)$. However, the degrees of inhibition for monophenolase ($i_{M}$) and diphenolase ($i_{D})$ activity for the compound trans-3,4 difluorocinnamic acid, since it is a competitive inhibitor, should be the same [13]. Furthermore, $i_{M}=68.6\pm 4.2\;\mu M$ and $i_{D}=780\pm 2\;\mu M$, and the problem stems from the fact that the inhibition study has been conducted through the measurement of monophenolase activity with L-tyrosine, so the lag is not eliminated and the speeds obtained may not be correct [27]. This compound essentially binds to the met form, according to the docking data (Table 2).

#### 2.3.2. With Reference to Non-Competitive Inhibitors

Non-competitive inhibition of tyrosinase by indazoles has been described (Appendix A) [28]. The values described for $K_{I}^{\mathrm{app}}$ are always greater than the IC_50_, and a possible explanation for this could be that the activity measurements are made with catechol, which derives a very unstable *o*-quinone after its oxidation. These compounds, according to docking studies, bind practically with the same affinity to the met and oxy forms (Table 2).

On the other hand, the study of tyrosinase inhibition by 2-aminobenzoic acid and 4-aminobenzoic acid shows strict compliance with Appendix A, and non-competitive inhibition [29] (Chemical structures are shown in Appendix A), obtaining IC_50_ values = 4.72 µM and 20 µM with values of $K_{I}^{\mathrm{app}}$ = 4.72 µM and 20 µM, respectively. The affinity of these compounds for the met form is much higher than for the oxy form, according to the docking data (Table 2). The presence of peroxide in the oxy-tyrosinase form makes the union of these compounds weaker (Table 2).

Inhibition of tyrosinase by hydroxyl-substituted benzoate/cinnamate derivatives has recently been reported (Figure 2) [34]. In this work, the IC_50_ values described do not correlate quantitatively with the $K_{I}^{\mathrm{app}}$ values described by Appendix A. Benzoate and cinnamate are competitive inhibitors, so these derivatives are expected to behave in the same way, but this is not the case, and they are described as non-competitive. A possible explanation for this could be that the same substrate concentration range is used for different inhibitor concentrations. In addition, these molecules have a free hydroxyl, which could be an enzyme substrate (Table 3 (see later)) which would distort the kinetic analysis (see Figure 2).

#### 2.3.3. With Reference to Uncompetitive Inhibitors

Inhibition of tyrosinase by a component of *Moringa oleifera* extract (Appendix A) has been described [35]. The type of inhibition described is uncompetitive, with an IC_50_ = 121.3 ± 0.4 µg/mL and $K_{I}^{\mathrm{app}}\;$= 73 µg/mL. As luteolin is the main component of the extract, these data would be in agreement with a study published a few years ago [38] where an inhibition constant value of 103 µM and an uncompetitive type of inhibition were proposed. However, luteolin has recently been described as a non-competitive inhibitor with $K_{I}^{\mathrm{app}}$ = 291.75 ± 7.75 µM [39]. These discrepancies of reversible uncompetitive and non-competitive inhibition can be explained by the chemical structure of luteolin (Appendix A), which, like any compound with a diphenolic structure, behaves as a substrate of tyrosinase [39]. The affinity of luteolin for met-tyrosinase is low (Table 3). Docking of luteolin to oxy-tyrosinse is shown in Appendix A and docking to met-tyrosinase in Appendix A. Because this molecule carries an *o*-diphenolic structure, it is oxidized by both met-tyrosinase and oxy-tyrosinase (Table 3).

#### 2.3.4. With Reference to Mixed-Type Inhibitors

The inhibition of tyrosinase by propylgallate is of the mixed type (Appendix A), with an IC_50_ value of 0.685 mM and with values of $K_{I_{1}}^{\mathrm{app}}$ = 0.661 mM and $K_{I_{2}}^{\mathrm{app}}$ = 2.135 mM. According to data from the docking (Table 2), this compound binds better to the met form than to the oxy [30]. The IC_50_ value that is obtained by applying the formula described in the Appendix A is 1 mM, and the value described experimentally is 0.685 mM.

### 2.4. Other Possible Causes of the Lack of Correlation between IC_50_ and $K_{I}^{app}$

#### 2.4.1. Inhibitor Can Be Alternative Substrate

Benzoate and cinnamate are competitive inhibitors of tyrosinase diphenolase activity [13]. In this sense, for the derivatives of benzoate and cinnamate studied in [34], a competitive type of behaviour should be expected, but what is described is a “non-competitive” behaviour. It is noteworthy that the quantitative relationship Appendix A between the values of IC_50_ and $K_{I}^{\mathrm{app}}$ is not fulfilled. Thus, (2-(3-methoxyphenoxy)-2-oxoethyl-2,4-dihydroxybenzoate) (compound a) had an IC_50_ value of 23.8 μM, while the $K_{I}^{\mathrm{app}}$ value was 130 μM. On the other hand, (2-(3-methoxyphenoxy)-2-oxoethyl-(E)-3-(4-hydroxyphenyl) acrylate) (compound b) inhibited tyrosinase with an IC_50_ value of 5.7 µM, while the $K_{I}^{\mathrm{app}}$ value was 11 µM. The structures of these compounds are shown in Figure 2, and it could be proposed that they may be substrates of the enzyme. This would result in obtaining incorrect initial rates.

The docking of benzoate and cinnamate to met-tyrosinase and oxy-tyrosinase is shown in Figure 3 and Figure 4. The docking of benzoate derivatives Figure 2a is shown in Figure 5 and that of cinnamate derivatives Figure 2b in Figure 6. From the data shown in Table 3, these compounds have practically the same affinity towards oxy-tyrosinase and met-tyrosinase and the distance of the oxygen from the peroxide in the oxy form to the ortho position with respect to the phenolic hydroxyl is 2.8 Ᾰ. This is adequate to produce the hydroxylation reaction, and this could be the origin of the change in the type of inhibition.

From the chemical structure of the two compounds shown in Figure 2a,b, a hydroxyl group can be seen in position 4. Docking studies in Figure 5 and Figure 6 show that the two compounds bind with high affinity for the hydroxyl in position 4, and also the distance of the oxygens of oxy-tyrosinase to the ortho position is adequate for the electrophilic aromatic substitution reaction to occur (Table 3) [40,41,42]. In this way, these compounds would possibly behave as alternative substrates to L-dopa. In the case of 4-hydroxycinnamic and 3-hydroxycinnamic, it was shown that they were alternative substrates to L-tyrosine and L-dopa [43,44]. The interactions with the hydroxyl group at positions 3 and 4 and the distances of the oxygens in oxy-tyrosinase to the ortho positions make hydroxylation possible [43,44].

#### 2.4.2. Inhibitor Can Be a Suicide Substrate

In general, *o*-diphenols are suicide substrates of tyrosinase with different inactivation potency. In Appendix A, the action of a possible inhibitor with an *o*-diphenolic group is shown, and this compound could be oxidized by both met-tyrosinase and oxy-tyrosinase. Furthermore, it should be noted that *o*-diphenols are suicide substrates of the enzyme. The same substrate used to measure L-dopa activity is also a suicide substrate, but the important thing is the time that the enzyme and *o*-diphenol are in contact. The suicidal action of L-dopa is minimal because the measurements are made at short times, but as in many inhibition assays, the enzyme is preincubated with the inhibitor (*o*-diphenol) from 10 to 30 min; the effect of suicide inactivation is relevant and this can lead to distortion of the kinetic analysis.

Appendix A shows the docking of compound (Z)-2-(3,4-Dihydroxybenzylidene)benzo[4,5]imidazo[2,1-b]thiazol-3(2H)-one to oxy-tyrosinase and met-tyrosinase. This compound can behave as a suicide substrate and cause deviations in the determination of kinetic parameters.

Appendix A show the docking of luteolin to oxy- and met-tyrosinase. Due to the *o*-diphenolic structure, this compound can be a substrate of the enzyme and have suicide behaviour [39].

In the case of propyl gallate (Appendix A), the IC_50_ value is 0.685 mM, lower than the theoretical value obtained from $K_{I_{1}}^{\mathrm{app}}$ and $K_{I_{2}}^{\mathrm{app}}$. This slight increase in the strength of the inhibitor may be because these trihydroxylated compounds are suicide substrates of tyrosinase, as was shown long ago [45,46].

Tyrosinase inhibition by four polyphenols from Morus and tulles Barley has been studied [31]. These compounds have been described as competitive inhibitors (Appendix A): sanggenone C (IC_50_ = 18.85 µM); L-epicatechin (IC_50_ = 191.99 µM); catechin (IC_50_ = 511.59 µM). Non-competitive inhibition has been reported for oxyresveratrol (IC_50_ = 4.50 µM). The $K_{I}^{\mathrm{app}}$ values described were: sanggenone C 11.92 µM; L-epicatechin 119.16 µM; catechin 365.86 µM and oxyresveratrol 4.50 µM. The assay methodology is similar to the cases described above with preincubations of the enzyme and the inhibitor dissolved in DMSO for 10 min. The reaction is started by adding L-dopa at a concentration equal to $K_{m}^{D}$, and the IC_50_ values should meet the relationship IC_50_ = $K_{I}^{\mathrm{app}}\left(1+n\right).$ With n = 1, these would be ${\mathrm{IC}}_{50}=2K_{I}^{\mathrm{app}}$; however, the IC_50_ values do not satisfy Appendix A. In the case of oxyresveratrol, it is true that IC_50_ $≅{\;K}_{I}^{\mathrm{app}}$, because it is a non-competitive type of inhibitor and could also be a substrate for the enzyme in the presence of H_2_O_2_ or L-dopa [47,48]. The docking results are shown in Table 3. In the case of sanggenone C, the compound acts as a competitive inhibitor (Table 2), acting on met-tyrosinase (Appendix A). The chemical structure of catechin and epicatechin show an *o*-diphenol, and they are substrates of the enzyme, as demonstrated experimentally [49]. The low IC_50_ values would be justified because these compounds are suicide substrates (Appendix A) of the enzyme and a 10 min preincubation is performed in the assay [49]. The values of the dissociation constants obtained from the docking are shown in Table 3. The docking Figures are described in Appendix A.

When the anti-melanogenesis and anti-tyrosinase power of phenethyl cinnamamides compounds from an extract of hemp (*Cannabis sativa* L.) was studied, it was found that among the compounds studied (Appendix A) [36], the most potent is the N-trans-caffeoyltyramine. Its diphenolic structure can cause the suicide inactivation of the enzyme, since in the inhibition test it is preincubated with tyrosinase for 30 min and then the reaction is started with L-dopa, in addition to an additional 10 min incubation. The three inhibitors, according to the docking data (Table 3), can be substrates of the enzyme, joining through the phenolic hydroxyl group corresponding to tyramine, in addition to compound (a), which due to its diphenolic structure, can be a suicide substrate. The dockings are shown in Appendix A. The distance of the oxygen from the peroxide to the ortho position of the phenolic hydroxyl is 2.8 Ᾰ and 2.9 Ᾰ, respectively, and therefore they can act as alternative substrates.

#### 2.4.3. Enzyme Inhibition Assay Design

In performing tyrosinase inhibition assays, several aspects must be considered in order to avoid false interpretations, as described below:

##### Preincubations and Use of Organic Solvents

It is common in tyrosinase inhibition assays to preincubate the enzyme with the inhibitor for a long time, and to subsequently start the reaction with the substrate. This procedure can give erroneous values of the inhibited rate ($V_{0,i}$) due to a possible reaction of the inhibitor with the enzyme [32]. Kinetic studies have been carried out on the inhibition and inactivation of tyrosinase by DMSO in the presence of substrate, and the inhibition constant of the free enzyme and of the enzyme–substrate complex has been determined. From these studies, it is shown that at low concentrations of DMSO, the inhibition is reversible, and at high concentrations, the enzyme is irreversibly inactivated. In addition, these works indicate that the substrate protects the enzyme from inactivation [50,51,52]. Inhibition of tyrosinase by cinnamate esters has been studied [32], showing that they are more potent than their parent compounds, see Appendix A [32]. The calculated values for the IC_50_ parameter are: 2.0 µM; 8.3 μM and 10.6 μM and the values of $K_{I}^{\mathrm{app}}$ and the type of inhibition were: (a) non-competitive and $K_{I}^{\mathrm{app}}$ = 3.8 μM; (b) mixed $K_{I}^{\mathrm{app}}$ = 10.0 μM and $K_{\mathrm{IS}}^{\mathrm{app}}$ = 35.6 μM (c) mixed and $K_{I}^{\mathrm{app}}$ = 8.0 μM and $K_{\mathrm{IS}}^{\mathrm{app}}$ = 72.2 μM. Theoretical IC_50_ values calculated from Appendix A are 15.6 μM and 7.20 μM. Note that the IC_50_ value for cinnamate is 209.5 µM, lower than that described in [13]. An explanation for this could be the inactivating effect of DMSO and the long incubation time. It is noteworthy that 4-hydroxycinnamic has a value of 4708.5 µM [32] as it is a substrate, which is in agreement with that described in [43]. Compounds (a) and (b) could behave as alternative substrates (Table 3); however, compound (c) is a true inhibitor (see Table 2). The dockings of (a) and (b) to Eox are shown in Appendix A.

Tyrosinase inhibition by two new stilbenes extracted from stems of *Streblus Ilicifolius* has been described. (Appendix A) [37]. Compound (a) bears a resorcinol end and is much more inhibitory than (b) which bears a phenol. On the other hand, the inhibition assay again involves the inhibitor dissolved in DMSO and a 30 min preincubation with which the DMSO can inactivate the enzyme. The reaction is subsequently started by adding L-dopa and is incubated again for 7 min. In our opinion, the design of the inhibition test is not adequate, and also, in the presence of L-dopa, these compounds could be alternative substrates. Compound (a) binds oxy better than (b) (Table 3). Figures of the docking of tyrosinase to these two compounds are shown in Appendix A. These compounds, especially streblus C, can behave as alternative substrates (Table 3).

### 2.5. Proposal for an Experimental Design to Determine the Type and Strength of a Tyrosinase Inhibitor from $i_{D}$ Values

An experimental design is proposed together with data analysis that allows several aspects to be determined with a few experiments—the IC_50_ value, the type of inhibition and the $K_{I}^{\mathrm{app}}$ value. This design is described and carried out with data obtained by simulating the different mechanisms under study. This would be carried out in the following stages:

Step 1. Progress curves in the absence and presence of inhibitor, obtained by simulation, according to the differential equations corresponding to each mechanism of inhibition, described in the Appendix A. Determination of the $i_{D}$ values, varying the concentration of inhibitor, at a fixed substrate concentration $\;{\left[D\right]}_{0}=K_{m}^{D}$ (Appendix A). Appendix A shows the data for the different types of inhibition considered in this paper.

Step 2. Representation of inhibition degree values versus inhibitor concentration. Figure 7 shows the values of $i_{D}$ vs. ${\left[I\right]}_{0}$ for each type of inhibition.

Step 3. Data analysis of $i_{D}$ vs. ${\left[I\right]}_{0}$ by nonlinear regression according to Appendix A. ${\mathrm{IC}}_{50}$ is determined for each inhibitor: competitive (${\mathrm{IC}}_{50}^{C}$); non-competitive (${\mathrm{IC}}_{50}^{\mathrm{NC}}$); uncompetitive (${\mathrm{IC}}_{50}^{U}$); mixed type (1) (${\mathrm{IC}}_{50}^{M_{1}}$) and mixed type (2) (${\mathrm{IC}}_{50}^{M_{2}}$) (Table 4).

Step 4. Determination of $i_{D}$ for ${\left[D\right]}_{0}=2K_{m}^{D}$ and with ${\left[I\right]}_{0}$ fixed (Appendix A).

Step 5. Possible types of inhibition:

$i_{D}^{2K_{m}^{D}}<i_{D}^{K_{m}^{D}}:$ could be competitive.$i_{D}^{2K_{m}^{D}}=i_{D}^{K_{m}^{D}}:$ is non-competitive and $K_{I}^{\mathrm{app}}={\mathrm{IC}}_{50}^{\mathrm{NC}}$.$i_{D}^{2K_{m}^{D}}>i_{D}^{K_{m}^{D}}:$ could be uncompetitive.$i_{D}^{2K_{m}^{D}}<i_{D}^{K_{m}^{D}}:$ ambiguity between competitive and mixed type (1).$i_{D}^{2K_{m}^{D}}>i_{D}^{K_{m}^{D}}:$ ambiguity between competitive and mixed type (2).

In case B, there is no ambiguity between the types of inhibition. It would be a non-competitive inhibitor with ${\mathrm{IC}}_{50}^{\mathrm{NC}}=K_{I}^{\mathrm{app}}$, from the value of ${\mathrm{IC}}_{50}^{\mathrm{NC}}$ (Table 4). The constancy of the value of $i_{D}$ when varying the concentration of the substrate, the type of inhibition (non-competitive) and the strength of the inhibitor are obtained for $K_{I}^{\mathrm{app}}$ (26.9 μM), as per Appendix A. In all other cases, there is ambiguity, which needs to be resolved.

Step 6. Solution of the ambiguity between competitive inhibition and mixed type (1). Two $i_{D}$ values are obtained by simulation with ${\left[D\right]}_{0}=5K_{m}^{D}$ and ${\left[D\right]}_{0}=10K_{m}^{D}$. If the inhibition decreases significantly (for example: from 25.77% to 16.11%) ($i_{D}\to 0$), then it would be a competitive inhibitor. If the inhibition decreases slightly (for example: from 51.02% to 48.37%), then it could be a mixed type (1) inhibition. In any case, the adjustment of the experimental data of $i_{D}$ vs. n according to Appendix A helps to discern between competitive inhibition and the mixed type (1) (Figure 8A,B). The inhibition constant ($K_{I_{1}}^{\mathrm{app}}$) is determined by nonlinear regression adjustment of ($i_{D}$ vs. n) according to Appendix A (Figure 8A). The determined value of $K_{I_{1}}^{\mathrm{app}}$ must comply with the value of ${\mathrm{IC}}_{50}^{C}$ according to Appendix A. For $K_{I_{1}}^{\mathrm{app}}$ (Table 4), in the case of competitive inhibition and in the case of mixed type (1) inhibition, the adjustment is performed by nonlinear regression of ($i_{D}$ vs. n) according to Appendix A, determining $K_{I_{1}}^{\mathrm{app}}$ and $K_{I_{2}}^{\mathrm{app}}$(Figure 8B). These values must comply with Appendix A and are described in Table 4.

Step 7. Solution of the ambiguity between uncompetitive and mixed type (2) inhibition. Figure 9A,B shows the values of $i_{D}$ obtained for the variation in ${\left[D\right]}_{0}$ at fixed inhibitor concentration.

## 3. Material and Methods

### 3.1. Enzyme Source

Mushroom tyrosinase (3130 U/mg) was purchased from Merck Life (Madrid, Spain) and purified as previously described [53]. Protein content was determined by Bradford’s method [54]. This enzyme is a tetramer with two heavy subunits, H, and two light ones, L [55].

### 3.2. Reagents

Benzoate, cinnamate and L-dopa were purchased from Merck Life (Madrid, Spain). Stock solutions of substrates were prepared in 0.15 mM phosphoric acid to prevent auto-oxidation.

### 3.3. Spectrophotometric Assays

Absorption was recorded in a visible-ultraviolet PerkinElmer Lambda 35-spectrophotometer (Perkin Elmer, Madrid, Spain), online interfaced with a compatible laptop. The temperature was maintained at 25 °C. Kinetics assays were also carried out with the above instruments by measuring the appearance of the products in the reaction medium. The activity on L-dopa was measured at 475 nm [56].

### 3.4. Simulation Assays

Initial rate ($V_{0}^{D,\mathrm{DC}}$) of the diphenolase activities in the absence (Figure 1) and presence of different concentrations of inhibitor (Figure 2) were calculated from the simulated progress curves obtained by numerical solution of the nonlinear set of differential equations corresponding to these mechanisms (see Appendix A). The systems of differential equations were solved numerically for particular sets of values of the rate constants and of initial concentrations of the species involved in the reaction mechanisms (WES) [57].

### 3.5. Kinetic Data Analysis

The initial velocity in the absence of inhibitor $V_{0}^{D,\mathrm{DC}}$, and in the presence $V_{o,i}^{D,\mathrm{DC}}$, were calculated by linear regression of the spectrophotometric recordings of the change in absorbance at 475 nm versus time. From these values, the degree of inhibition is determined:$$i_{D}\%=\left(V_{0}^{D,\mathrm{DC}}-V_{o,i}^{D,\mathrm{DC}}/V_{0}^{D,\mathrm{DC}}\right)\times 100$$

Non-linear regression analysis [58] of the $i_{D}\%$ values with respect to ${\left[I\right]}_{0}$ obtained at the same substrate concentration allows for obtaining the IC_50_. For each type of inhibition there is a relationship between IC_50_ and the inhibition constant (competitive, non-competitive and uncompetitive) or constants (mixed). When the substrate concentration is varied, keeping the inhibitor concentration fixed, the non-linear regression analysis of the corresponding equation provides the values of $K_{I_{1}}^{\mathrm{app}}$: competitive inhibition, Appendix A; $K_{I_{2}}^{\mathrm{app}}$: uncompetitive inhibition, Appendix A; and both constant at mixed inhibition, Appendix A.

The REFERASS computer program was used to obtain the rate equations of these mechanisms in the presence of inhibitor or of an alternative substrate [59].

### 3.6. Computational Docking

Molecular docking of the ligands was studied in the active site of mushroom tyrosinase. Their chemical structures were built from chemical structures obtained from the PubChem Substance and Compound database [60] (Appendix A). The molecular structure of tyrosinase was taken from the Protein Databank (PDB ID:2Y9W, Chain A) [55], corresponding to the deoxy-form of tyrosinase from *Agaricus bisporus*. Input protein structure for docking was prepared by adding all hydrogen atoms and removing water molecules. The met and oxy forms of tyrosinase were built as previously described [61]. Gasteiger’s partial charges and rotatable bonds were assigned by AutoDockTools4 software [62,63]. AutoDock 4.2.6 software package [63] was used for docking calculation. Lamarkian Genetic Algorithm was chosen to explore the space of active binding to search for the best conformers. Grid parameter files were built using AutoGrid 4.2.6 [64]. Other docking parameters were used as in [65]. PyMOL 2.3.0 (Schrödinger) was employed to build and inspect the molecule structures and docked conformations [66]. Docking conformations were selected from clusters of conformations that can lead to ligand catalysis with the lowest free energy of binding. Ligand–protein interactions were analysed using PLIP software [67].

## 4. Conclusions

The study of tyrosinase inhibition is very important due to the multiple applications it can have in pathological processes of pigmentation, such as the browning of fruits and vegetables. The importance of the IC_50_ parameter and its quantitative relationships with the apparent inhibition constants are highlighted. Thus, from the IC_50_ value, the inhibition constant for a competitive, non-competitive, or uncompetitive inhibitor can be determined when the inhibition mechanism is confirmed. In the case of mixed inhibition, the $i_{D}$ value obtained at different substrate concentrations is adjusted against n ($n={\left[D\right]}_{0}/K_{m}^{D}$). Enzyme inhibition assays are discussed and some techniques for their optimization are proposed.

## Data Availability

Not applicable.

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
