# Peer review of "The Relationship between the IC50 Values and the Apparent Inhibition Constant in the Study of Inhibitors of Tyrosinase Diphenolase Activity Helps Confirm the Mechanism of Inhibition"

_molecules, 2022, doi:10.3390/molecules27103141_

Round 1
Reviewer 1 Report
See attached file

Author Response
We welcome the referee's comments and suggestions. These suggestions from the Referee help to improve the work and we have followed them and corrected aspects that, thanks to the Referee, have been revealed. He then went on to respond point by point to the various comments.
General comments.
1) This paper scrutinizes how the inhibition of a tyrosinase can be analyzed using enzyme kinetic assays and determination of IC50 values. It points out some challenges related to the complexities of the catalyzed reaction as well as how the reactions are influenced by various experimental conditions and that inhibitors may in fact be alternative substrates or inactivators. Examples from published work illustrates that these issues have often not been properly addressed.
We agree with the Referee that in this work we want to highlight the value of the IC50 parameter, determined from the dependence of on the inhibitor concentration, and apply this parameter to the study of different inhibition mechanisms. The biggest problem arises with the complex mechanism of the enzyme tyrosinase. In this work it is highlighted that some apparent inhibitors are alternative substrates or suicide substrates. The purpose of this work is to highlight this aspect, since tyrosinase inhibitors are being sought that may have applications in the clinic, such as the case of hydroquinone, which has been used very frequently as a depigmenting agent but has been shown to under certain conditions it can be a substrate for the enzyme.
2) The authors propose how experiments can be designed and analysed in order for reliable quantification of inhibition, based on different types of inhibition mechanisms. However, this is not clearly described, but very confusing and it is impossible to figure out how this should be done in practice. It appears to be a combination of simulations and experiments.
Certainly, the work tries to study the different mechanisms of action of inhibitors. There is an abundant bibliography of the different types of inhibition. We regret that we have not been able to establish a clear methodology. We believe that in the corrected version many aspects indicated by the Referee have been clarified.
Regarding that the methodology could be a combination of simulations and experiments, we have to say no, and we are sorry that we have given rise to that confusion.
The proposed methodology is experimental. A first totally experimental case is analyzed: benzoate and cinnamate as tyrosinase inhibitors. We studied these inhibitors in a previous work [1], the type of inhibition and the values ​​were determined. In this work the IC50 are determined from the values ​​at different concentrations of inhibitor and fixed substrate concentration, confirming that cinnamate is a more powerful inhibitor than benzoate: .
In addition, it is fulfilled from the dependence of IC50 with the substrate concentration, which is of the competitive type and is calculated (Table 1).
The simulation has been carried out in section 2.5. Proposal for an experimental design, which we now determine: "Proposal for an experimental design to determine the type and strength of a tyrosinase inhibitor from values" and a series of steps are proposed that would allow the quantitative analysis of an inhibitor.
3) Part of the problem with the paper is that the conventional terminology defining inhibition types (competitive, non-competitive and un-competitive) do not account for the complexities of tyrosinase catalysis. It is therefore not clear what these terms stand for with respect to the reaction scheme. The terminology is consequently not very helpful for the type of complexity exhibited by the enzyme studied and the new type of analysis is therefore not really helpful.
It is necessary to adjust the terminology in the analysis so that it is clear to which molecular species the inhibitor can bind, and what the affinity would be to this (or these) species. If not, this becomes a very academic exercise that provides few advantages over a pragmatic determination of IC50-values.
We agree with the Referee that the catalytic mechanism of this enzyme is complex, but we consider the types of inhibition that normally occur to enzymes in catalytic processes, in fact these types of inhibition are those described in different Review [2–8].
4) The start of the paper says “In the study of the inhibition of the diphenolase activity of tyrosinase (a trisubstrate enzyme), it is interesting to know the strength and type of inhibition.” But this is actually not well addressed in the paper since the “type of inhibition” is more complex than the standard types that can be used for a simple system (one substrate and a reaction scheme with only on free enzyme species and only one species of substrate bound enzyme). Also, the goals of the paper rows 76-85 are not met with the work described.
We agree with the Referee that the objective of this work and how to carry it out may not have been clearly described. Following the Referee's suggestions both in the Abstract and in lines 76-95, the usefulness of the IC50 parameter has been clarified, the most used in reviews [2–8], and the possibility of analyzing an inhibitor from the determination of IC50 and in relation to substrate concentration.
Specific comments (not comprehensive)
Abstract
Add a comment on the relevance of the work and why it is valuable to the field. Need to explain the challenge of studying an enzyme that catalyses two sequential reactions and that the second reaction (described in the paper) can be seen as different steps with different intermediates to which an inhibitor can bind (only one or several, and which ones), as shown in Scheme II. This is more relevant for the context than explaining that it is a trisubstrate enzyme.
The study of tyrosinase inhibition is very important because it is about solving pigmentation pathologies due to an abnormal action of the enzyme and trying to control its activity through the use of inhibitors.
Row 27: “… for type inhibitors” is not clear, something missing?
This sentence has been removed in the new version.
Introduction
An explanation for why it is relevant to establish the type of inhibition is lacking and how it provides information of the relevance for researchers interested in inhibiting the enzyme.
It has been added following the instructions of the Referee because it is relevant to establish the type of inhibition. If the type of inhibition is known, we can have an analytical expression that allows us to calculate the of an inhibitor under study that allows us to compare it quantitatively with another. This concept has been added in the introduction.
Results and discussion
Row 354-356: It is not clear why organic solvents inhibit tyrosinase, it is stated as a fact. The sentence needs clarification.
The sentence has been changed to: “Kinetic studies have been carried out on the inhibition and inactivation of tyrosinase by DMSO in the presence of substrate and the inhibition constant of the free enzyme and of the enzyme-substrate complex has been determined. From these studies, it is shown that at low concentrations of DMSO, the inhibition is reversible and at high concentrations, the enzyme is irreversibly inactivated. In addition, these works indicate that the substrate protects the enzyme from inactivation [9–11]”
Row 409- : The section on the proposal of an experimental design is not clear. Why start with a simulation?
The beginning of this section has been changed, but we want to indicate that the experimental design is made visible with a simulated example with our WES program [12].
Row 419: It seems that the figure legend and the main text have become mixed up, it is not possible to decipher.
The Referee is right, in the new version the main text of Figures 8A, B and 9A, B has been separated.
Row 440: What data is to be analysed? The simulated data?
The data analyzed in this example is simulated to drive the experimental design and corresponding data analysis. In the case of the experimental study of an inhibitor, a similar method had to be followed.
Row 484: Here also the main text and figure legend seem mixed up
In this case, the main text of the Figure has also been separated, following the instructions of the Referee.
Minor issues
Layout: IC50 should be written with “50” as subscript in the title and abstract. The abstract also has confusing typography of constants. There are some typos/linguistic errors that need to be fixed.
IC50 expressions have been corrected.
It should be clarified that the experiments have been done with mushroom tyrosinase and what the structural features are for this enzyme (H2L2 tetramer).
It has been added that the experiments have been done with mushroom tyrosinase, which is a tetramer with two heavy subunits, H, and two light ones, L [13].
Row 361-369: The IC50-and Ki values are expressed in units of μg/ml, it would be better if they were converted to M since they are being compared to each other.
In these rows, the IC50 values ​​have been changed from units of μg/mL to μM to be comparative as indicated by the Referee.
Reviewer 2 Report
It is an interesting study. Please correct typos and make it understandable as easily as possible.
line 53, 'nona' to 'none'
line 60, delete 'KIapp'
Line 60, 'amount' to 'concentration'
line 62, 'the inhibition constant' to 'the apparent inhibition constant'
line 64, delete 'the apparent inhibition constant'
line 66, 'acompetitive' to 'uncompetitive'
line 67, define iD
line 70, 'the limiting rate' to 'the maximal rate'
line 78, define [D]0
Please consider using DQ' for 'dopaquionone'.
Please use one of benzoic acid or benzoate, consistently.
Please use one of cinnamic acid or cinnamate, consistently.
In Fig 1, define iD and VoD,DC, Vo,i,D,DC.
Table 1, Move foot notes to under the table.
Figure 2, Chemical structure: Benzoate (A), cinnamate (B), and their derivatives (a,b).
Table 2 and 3, Delete 'Docking to oxy and met-tyrosinase'
Table 3, 'o-diphenol' to 'o-Diphenol'
Step 2, Please cite Figure 6 and separate Figure 6.
Please cite Figure 7.
Step 7, Please cite Figure 8 and separate Figure 8.
Author Response
Referee #2:
We appreciate the comments and suggestions of Referee 2, which have been taken into account in the new version.
Comments and Suggestions for Authors
It is an interesting study. Please correct typos and make it understandable as easily as possible.
1) line 53, 'nona' to 'none'.
The indicated change has been made.
2) line 60, delete 'KIapp'.
Kiapp has been removed at line 60.
3) Line 60, 'amount' to 'concentration'.
Following the instructions of the Referee, amount has been changed to concentration.
4) line 62, 'the inhibition constant' to 'the apparent inhibition constant'.
The first sentence has been exchanged for the second.
5) line 64, delete 'the apparent inhibition constant'.
The apparent inhibition constant has been deleted in line 64.
6) line 66, 'acompetitive' to 'uncompetitive'.
The change indicated by the Referee has been made.
7) line 67, define iD.
has been defined as indicated by the Referee.
8) line 70, 'the limiting rate' to 'the maximal rate'.
First sentence has been modified by the second as indicated by the Referee.
9) line 78, define [D]0.
The term has been defined as indicated by the Referee.
10) Please consider using DQ' for 'dopaquionone'.
Dopaquinone has been changed to DQ as indicated by the Referee.
11) Please use one of benzoic acid or benzoate, consistently.
Benzoate has been chosen and homogenously used throughout the paper.
12) Please use one of cinnamic acid or cinnamate, consistently.
Cinnamate has been used throughout the paper.
13) In Fig 1, define iD and VoD,DC, Vo,i,D,DC.
Following the suggestions of the Referee, these terms have been defined.
14) Table 1, Move foot notes to under the table.
The change has been made in Table 1.
15) Figure 2, Chemical structure: Benzoate (A), cinnamate (B), and their derivatives (a,b).
The change indicated by the Referee has been made.
16) Table 2 and 3, Delete 'Docking to oxy and met-tyrosinase'.
What was indicated by the Referee has been eliminated.
17) Table 3, 'o-diphenol' to 'o-Diphenol'.
o-diphenol has been changed to o-Diphenol.
18) Step 2, Please cite Figure 6 and separate Figure 6.
It has been changed to follow the Referee's suggestions.
19) Please cite Figure 7.
Figure 7 has been cited.
20) Step 7, Please cite Figure 8 and separate Figure 8.
The change indicated by the Referee has been carried out.
Reviewer 3 Report
- In general, the narrative text is too long, and difficult to grasp the key points. A focus on reducing redundancy would be recommended.
- Page 2, Line 44. I would suggest adding the metabolic pathway for a clear demonstration.
- Regarding Docking results, take Figure 3 for example.
- Figure 3 is the first figure to show the docking results. Please show the whole tyrosinase enzyme. We can see the active site and binding position.
- Analyze and show the interaction of ligand and enzyme, such as H-bonding, by using the LigPlot+, UCSF Chimera X, etc. Comparing the chemical interactions between different types of inhibitions
- Please describe the scoring function in the Docking studies. Show the scoring ranking for the ligands (compounds) in this study, and evaluate the relationship between ranking and the inhibition types
Author Response
We appreciate the Referee's suggestions and we proceed to answer him point by point.
Comments and Suggestions for Authors
1) In general, the narrative text is too long, and difficult to grasp the key points. A focus on reducing redundancy would be recommended.
Following the suggestions of the Referee, in the new version the key points of the work have been highlighted and an attempt has been made to eliminate redundancy. Two Figures and two references have been removed.
2) Page 2, Line 44. I would suggest adding the metabolic pathway for a clear demonstration.
Scheme I. Melanin biosynthesis pathway. M, monophenol (L-tyrosine); D, o-diphenol (L-dopa); DQ, (L-o-dopaquinone); L, leucodopachrome and DC, L-dopachrome.
3) Regarding Docking results, take Figure 3 for example.
- Figure 3 is the first figure to show the docking results. Please show the whole tyrosinase enzyme. We can see the active site and binding position.
The whole tyrosinase structure is now depicted in new Figures 3 and 4.
- Analyze and show the interaction of ligand and enzyme, such as H-bonding, by using the LigPlot+, UCSF Chimera X, etc. Comparing the chemical interactions between different types of inhibitions
Analysis of the interactions of all docking positions shown in the manuscript has been detailed in Table S4 in the supplementary material.
- Please describe the scoring function in the Docking studies. Show the scoring ranking for the ligands (compounds) in this study, and evaluate the relationship between ranking and the inhibition types
Docking conformations correspond only for the cases of competitive inhibition where ligands can also be substrates. Thus, the criterion for the choice of docking conformations was to search for cluster of conformations that can lead to competitive inhibition with the lowest free energy of binding. Thus, docking was studied only in the active site of tyrosinose not in the whole structure of the protein. These issue has now been stated in Method section [14].